# JP2 Genotype of *Aggregatibacter actinomycetemcomitans* in Caucasian Patients: A Presentation of Two Cases

**DOI:** 10.3390/pathogens9030178

**Published:** 2020-03-01

**Authors:** Alexandra Stähli, Anton Sculean, Sigrun Eick

**Affiliations:** Department of Periodontology, School of Dental Medicine, University of Bern, Freiburgstrasse 7, 3010 Bern, Switzerland; anton.sculean@zmk.unibe.ch (A.S.); sigrun.eick@zmk.unibe.ch (S.E.)

**Keywords:** JP2 clone of *Aggregatibacter actinomycetemcomitans*, periodontitis, JP2 in Caucasian, microbiological diagnosis, adjunctive antibiotics

## Abstract

*Aggregatibacter actinomycetemcomitans* is a key pathogen that has been associated with periodontal disease. Its most important virulence factor is a leukotoxin capable of inactivating immune cells. The JP2 genotype of *Aggregatibacter actinomycetemcomitans* shows enhanced leukotoxic activity and is mostly present in individuals of North and West African origin with severe periodontitis. In this paper, two cases of Caucasians diagnosed with the JP2 genotype are presented. A 50-year-old female patient had three approximal sites with ≥ 6 mm clinical attachment loss (CAL) and eight sites with probing depth (PD) ≥ 5 mm. Microbiological diagnostics revealed *A. actinomycetemcomitans* JP2 genotype, but not *Porphyromonas gingivalis*. This JP2 genotype was highly leukotoxic to monocytic cells. The second case was a 55-year-old female patient with CAL of > 5 mm at all molars and PD of up to 12 mm. *A. actinomycetemcomitans* JP2 was identified, but not *P. gingivalis*. Her husband originated from North-Africa. In him, no *A. actinomycetemcomitans* was detected, but their 17-year-old daughter was diagnosed with periodontitis and was found to be positive for the JP2 genotype. Both patients were successfully treated with adjunctive antibiotics and the JP2 genotype was eliminated. In summary, here, the microbiological diagnosis was key for the treatment with adjunctive antibiotics.

## 1. Introduction

Bacterial biofilm causes destruction of the periodontium in two ways: through direct action of bacteria and their products on the host-tissue and by activating the immune host response [1]. *Aggregatibacter actinomycetemcomitans* is one of the key pathogens in the course of periodontal disease. *A. actinomycetemcomitans* has been strongly associated with localized aggressive periodontitis [2], however, its mere presence could not be used to distinguish between chronic (CP) and aggressive forms of periodontitis (AP) [3]. In the Department of Periodontology, School of Dental Medicine, University of Bern, patients diagnosed with AP generally received antibiotics during nonsurgical periodontal therapy (i.e., hygienic phase). Retrospective analysis of our patients revealed that the prevalence of *A. actinomycetemcomitans* was higher in patients diagnosed with AP than in those diagnosed with CP [4]. Following periodontal therapy, especially surgical treatment, *A. actinomycetemcomitans* was less frequently detected in patients with AP than in those with CP [4]. *A. actinomycetemcomitans* possesses several virulence factors, that is, lipopolysaccharides that induce pro-inflammatory cytokines, a cytolethal distending toxin causing cell cycle arrest in T-cells, macrophages and epithelial cells, and a leukotoxin [5]. The leukotoxin produced by the bacterium is capable of killing or inactivating immune cells and of inducing the release of interleukin (IL)-1β [6].

Here, we focus on a subtype of *A. actinomycetemcomitans*, the highly leukotoxic JP2 genotype, which was first isolated from a child of African American origin with prepubertal periodontitis [7]. Later, it was found as a common isolate in individuals of North and West African descent with aggressive forms of periodontitis [8,9,10,11,12]. With respect to the JP2 genotype, a 530 base pair deletion in the promoter region of the leukotoxin gene is responsible for a 10- to 20-fold increased production of leukotoxin [13,14]. The JP2 clone is a subpopulation of the serotype b strains [15].

To date, seven serotypes of *A. actinomycetemcomitans*, designated from a to g, have been identified [16]. Among them, serotypes a, b, and c are globally dominant, whereby type c is the most prevalent [17]. Interestingly, they show different associations with disease depending on ethnicity, geographical localization, or periodontal status. For example, in the United States, serotype c was mostly associated with AP, but also other strains, and the JP2 genotype was found in patients suffering from periodontitis [18]. In Brazil, serotype c was found to be the most prevalent one and associated with both AP and CP. On the other hand, serotype b was also detected in periodontally healthy individuals [19]. Conversely, others found a connection between serotype b and aggressive periodontitis [20]. In Japan, serotype c was predominantly isolated from patients with AP, while the occurrence of serotype b was rare [21]. The specific JP2 genotype of *A. actinomycetemcomitans* was found to be strongly associated with severe periodontitis, particularly in Northern and Western Africa [22,23]. In Asia, the occurrence of the JP2 genotype has not been reported so far [23], and in Germany, it was detected in immigrants from North Africa living for more than 10 years in Germany, but not in Caucasians [24]. Dissemination of the JP2 genotype to non-African populations was only very rarely described [22]. Nevertheless, recent data obtained from nearly 3500 subgingival plaque samples of 1445 periodontitis patients in Sweden showed that the JP2 genotype was found in 1.2% of patients and most of them were of non-African descent [25]. Furthermore, serotype b was more often found in younger patients with periodontitis than in older cohorts [25]. 

In our department, microbiological diagnostics of subgingival biofilm samples is routinely performed. This includes subtyping of *A. actinomycetemcomitans* strains. After identifying the JP2 genotype in an immigrant from Morocco in 2013, such a clone was detected in two periodontitis patients of non-African origin. Here, the two cases starting with the diagnosis together with all steps of periodontal treatment are presented.

## 2. Results

### 2.1. Case 1

A 50-year-old female patient presented with localized CP according to the Classification System for Periodontal Diseases and Conditions set in 1999 [26]. At the initial examination, the patient was diagnosed with severe CP with three approximal sites with clinical attachment loss (CAL) ≥ 6 mm and eight sites with probing depth (PD) ≥ 5 mm (Table 1), as defined by the Centre for Disease Control and Prevention and the American Academy of Periodontology (CDC–AAP) [27,28]. No furcation involvement was detected. The patient reported to smoke occasionally. She was healthy and took no medications. Besides the third molars, teeth 16 and 27 were missing. The endodontically treated tooth 47 was scheduled for extraction because of an apical osteolysis. Besides this, the radiographs showed no further pathologies. No angular bony defects were visible and slight horizontal bone loss was noted. Microbiological diagnostics revealed high counts (more than 10^5^) for *A. actinomycetemcomitans*, low counts (about 10^4^) each for *Tannerella forsythia* and *Treponema denticola*, but no *Porphyromonas gingivalis*. Subtyping of *A. actinomycetemcomitans* showed a serotype b strain being positive for the deletion in the promotor region of in the leukotoxin operon (JP2 genotype). Another *A. actinomycetemcomitans* (without deletion in the promoter region of the leukotoxin operon) was not detected. At the next visit, she was asked for contact with people from North and Western Africa, but she had never been abroad before, nor had she closer contact to Africans. Further, additional biofilm was sampled to confirm the microbiological analysis and to culture the strain. Cultivation confirmed the high counts (10^5^) and identification *of A. actinomycetemcomitans*. Determination of antibiotic resistance found a minimal inhibitory concentration (MIC) of ≤ 0.5 µg/mL for amoxicillin and 4 µg/mL for metronidazole. The MTT (3-(4,5-dimethylthiazol-2-yl)-2,5-diphenyltetrazolium bromide) tetrazolium assay confirmed a very high toxicity of that strain, being remarkably higher than those of the control JP2 genotype reference strain (HK1651) (Figure 1). Meanwhile, the patient improved her oral hygiene, and no *T. forsythia* and *T. denticola* were found anymore, but *A. actinomycetemcomitans* was still present in high counts. In order to eradicate *A. actinomycetemcomitans*, the further treatment plan entailed a hygienic phase with antibiotics (amoxicillin 375 mg and metronidazole 500 mg each tid for seven days). After the initial oral hygiene instruction and supragingival scaling, the patient showed good oral hygiene with a plaque index (O‘Leary [29]) of < 20% of all tooth surfaces. Thereafter, subgingival scaling was performed in one session by hand curettes and an ultrasonic device with local anesthesia. Antibiotics as mentioned before and 0.02% chlorhexidine were given for 7 and 14 days, respectively. Tooth 47 was extracted. Three months after nonsurgical therapy, the patient was re-evaluated. The goals of periodontal therapy were achieved in all quadrants. There was no site with PD ≥ 5 mm and, therefore, no further surgical periodontal therapy was needed. Microbiological testing revealed an absence of *A. actinomycetemcomitans*, *P. gingivalis*, *T. forsythia*, and *T. denticola*. For supportive periodontal therapy, the patient was sent back to her dentist in private practice.

### 2.2. Case 2

A 55-year-old female patient was referred to the clinic for periodontal treatment after having been diagnosed with chronic generalized periodontitis. The periodontal screening index [30] was 4 for each sextant and the periodontal chart showed CAL of > 5 mm at all molars. At teeth 11 and 22, PDs up to 12 mm were detected. In the maxilla, all molars showed a furcation involvement degree II at least at one side. Teeth numbers 18, 38, and 48 were missing. In the mandible, all molars showed degree I furcation involvement. The patient was systemically healthy and a non-smoker. On the radiographs, horizontal bone loss was detected at the distal aspects of teeth 16, 15, 25, and 26. Angular bony defects were observed mesially of all first molars as well as distally of tooth 11. Microbiological analysis revealed high counts (more than 10^5^) for *A. actinomycetemcomitans*, low counts (about 10^4^) each for *T. forsythia*, and moderate counts (about 10^5^) for *T. denticola*, but no *P. gingivalis*. Subtyping of *A. actinomycetemcomitans* showed a serotype b strain being positive for the deletion in the promotor region of the leukotoxin operon (JP2 genotype). No other *A. actinomycetemcomitans* strain was identified. Her strain could be cultured (about 10^5^ per sample) and showed low MIC values to amoxicillin (≤ 0.5 µg/mL) and a resistance to metronidazole (32 µg/mL). The toxicity of the strain to the MONO-MAC-6 cells was similar to that of the JP2 genotype reference strain (HK1651), but also higher than those of the Y4 strain (serotype b strain without deletion in the promotor region) (Figure 1).

At the next visit, she was asked for contacts with Africans and, indeed, she was married to a man from North Africa. Her husband agreed to a periodontal clinical diagnosis including a microbiological analysis. However, he was periodontally healthy and no *A. actinomycetemcomitans* was detected. In the following, her children also agreed to a periodontal clinical and microbiological diagnosis. The 17-year-old daughter was diagnosed with aggressive periodontitis together with a positive detection for *A. actinomycetemcomitans*, but no *P. gingivalis*. After this accidental diagnosis, the daughter also received periodontal therapy, including adjunctive antibiotics.

After a thorough oral hygiene instruction and supragingival scaling, the patient showed an excellent oral hygiene and a plaque index (O‘Leary [29]) of less than 15%, and subgingival scaling was performed in two sessions within one week using hand curettes and local anesthesia. Upon the second session, antibiotics (amoxicillin 375 mg and metronidazole 500 mg each tid for seven days) were given because of the JP2 genotype detection and the severity of the tissue destruction. The reevaluation showed a conspicuous improvement with residual PD > 5 mm at teeth 11, 46, and 36. The latter further improved until the first recall so that surgical therapy was needed only for tooth 11 distally with PD of 9 mm and 46 mesially with PD of 7 mm. Now, the JP2 genotype of *A. actinomycetemcomitans* could not be detected anymore. For both teeth, a flap was raised using the simplified papilla preservation technique. After removal of granulation tissue, scaling and root planing was performed. Tooth 11 exhibited a three-wall defect with an intraosseous depth of 3.5 mm. Given the defect configuration, enamel matrix derivative and bone graft material were administered into the defect. Tooth 46 mesially exhibited a narrow angular bony defect of 3 mm depth, which was treated by means of an access flap surgery and application of an enamel matrix derivative. The patient was enrolled into a three-month recall at the Department of Periodontology. After one year, the recall interval was reduced to six months. A stable periodontal situation was noted with no PD > 4 mm. After another year, the patient was sent back to her dentist in private practice.

## 3. Discussion

In the present paper, we reported on two rare cases of *A. actinomycetemcomitans* JP2 genotype infection in Caucasians, highlighting the treatment sequences, the clinical outcomes, and the potential value of microbiological testing for the early detection of periodontal disease. Although, over the last 10 years, each *A. actinomycetemcomitans* positive sample has been screened for JP2 genotype presence, no further cases have been detected in our clinic up to now. This is in line with the findings of others who have only sporadically reported on the detection of JP2 genotype in non-African populations [31]. Conversely, the JP2 genotype is widespread and highly present in Northwest African populations. The reason this genotype of *A. actinomycetemcomitans* has remained geographically restricted despite globalization is still an unanswered question. However, it cannot be excluded that a specific host tropism exists that favors the colonization among these populations.

The highly leukotoxic JP2 genotype is strongly associated with AP. In Northwest African countries, there is a higher prevalence of AP reported among the young population than in other parts of the world, where it is a rare disease with a prevalence of less than 1% [32,33]. In contrast, both patients presented here were, at the time of the baseline examination, between 50 and 55 years of age and diagnosed with CP. An association between young age and the presence of JP2 genotype has been observed, however, with the increasing age of the host, these strains seem to disappear [12]. A prospective longitudinal cohort study has demonstrated that, initially, periodontally healthy subjects harboring the JP2 clone are more likely to develop periodontal attachment loss; a much less pronounced disease risk was found for those not carrying the JP2 genotype [33]. In our cases, it is unclear at what age the patients were infected by the JP2 genotype and how fast the periodontal defects evolved. The 50-year-old patient had no association with North or West African countries. The 55-year-old patient was married to a North African man.

It is of interest to note that there was no detection of *P. gingivalis* in any of these two cases. The fact that *P. gingivalis* was not detected could be correlated with the ability of certain subgingival bacteria to modulate the leukotoxicity of *A*. *actinomycetemcomitans.* Antibodies raised against *A*. *actinomycetemcomitans* and its leukotoxin may be inactivated by proteases of other bacteria such as *P. gingivalis* [34]. Gingipains are the primary virulence factor of *P. gingivalis*, showing a proteolytic activity against a broad spectrum of proteins [35]. Further, it has been shown that leukotoxin is proteolytically degraded by the action of gingipains [34]. *P. gingivalis* was able to completely destroy the leukotoxin of *A*. *actinomycetemcomitans* within an hour [34].

*A*. *actinomycetemcomitans* leukotoxin affects immune cells to release IL-1β [6]. Here, we tested the toxicity of the JP2 genotype strains of the two cases on MONO-MAC-6 cells. The JP2 genotype reference strain (HK1651) was more toxic than the Y4 strain. The difference might be not very high, but can be related to the experimental conditions using a lower bacterial concentration and a different cell line than that reported before [36]. One JP2 genotype strain showed a similar cell toxicity to the JP2 genotype reference strain on MONO-MAC-6 cells. The patient with that strain showed periodontal defects at the molar region and at the maxillary incisors, reflecting the typical localization pattern of AP [37].

However, MTT cytotoxicity assay revealed strong cytotoxicity of the other strain, interestingly, the case with no contact to Africans. Here, we can only speculate if there is a difference in production of leukotoxin or in retaining it at the cell surface. Leukotoxin is enriched in outer-membrane-like vesicles [36]. The obviously very high toxicity may contribute to the infection of a person with no genetic predisposition. Here, it has to be pointed out that leukotoxicity may be different depending on the test method that was used [38]. Variation of leukotoxicity was not only observed among JP2- and non-JP2 genotype of *A. actinomycetemcomitans,* but also among the methods of Western blotting, ELISA, cell lysis assay, and mRNA expression assay [38]. 

After diagnosis, the patients received two sessions of oral hygiene instructions and supragingival scaling by means of ultrasonic and hand instruments. Thereafter, subgingival non-surgical instrumentation was performed with adjunctive antibiotic therapy (amoxicillin and metronidazole). Microbiological testing after the hygienic phase showed that *A. actinomycetemcomitans* was no longer detected. In these cases, the microbiological characterization of *A. actinomycetemcomitans* strains influenced the therapeutic approach, namely to administer or not adjunctive systemic antibiotics. Otherwise, in view of the increasing bacterial resistance, patients diagnosed with CP are not treated with adjunctive amoxicillin and metronidazole during non-surgical mechanical therapy. The in vitro resistance of the strains to amoxicillin and metronidazole was determined. These data cannot be transferred directly to the clinic. A synergism between metronidazole and amoxicillin is well known, as amoxicillin increases the uptake of metronidazole in the bacterial cells [39]. First, this combination was successfully used to treat patients with *A. actinomycetemcomitans* associated periodontitis [40]. Nonetheless, it is well documented that, in patients diagnosed with CP or AP, better clinical outcomes can be obtained if systemic antibiotics are administered in conjunction with subgingival mechanical debridement, irrespective of their microbiological profile [41]. Therefore, in general, microbiological testing was found to be clinically irrelevant for the treatment strategy. It was demonstrated that the presence of putative periodontal pathogens quantified before the treatment was not key for the outcome of scaling and root planing (SRP) with or without amoxicillin and metronidazole [42].

A study evaluating the treatment response of patients infected with JP2 or non-JP2 genotype of *A. actinomycetemcomitans* has shown that patients infected with JP2 genotype had higher PD, CAL, and gingival inflammation than those infected with non-JP2 genotype at baseline. Patients with persisting JP2 genotype after full-mouth SRP and adjunctive administration of amoxicillin and metronidazole had increased gingival inflammation compared with patients where the JP2 strain was eliminated [43]. In the non-JP2 genotype-infected group, the clinical improvements in terms of PD reduction and CAL gain were statistically significantly higher compared with patients infected with the JP2 genotype. These data appear to suggest that the persistence of JP2 genotype in periodontal pocket diminishes the treatment response, which in turn may favor the progression of periodontitis.

In our material, we presented two cases of Caucasians infected with JP2 genotype who were successfully treated with full-mouth SRP and amoxicillin and metronidazole. The microbiological diagnosis was the key decision making factor for selecting the treatment strategy, including the use of amoxicillin and metronidazole. Additionally, it is important to point out that the microbiological diagnosis has finally led to a screening of the patients’ family members and the diagnosis of an AP in the teenage daughter of one patient.

## 4. Materials and Methods 

The two patients of non-African origin with detection of the JP2 genotype underwent active periodontal therapy at the Department of Periodontology during the years 2014 and 2015. Both patients were diagnosed with CP according to the classification set in 1999 [26]. The severity and extent of periodontal destruction varied considerably. For each patient, pooled samples of the deepest pockets of each quadrant were analyzed for the major bacteria associated with periodontal diseases using nucleic acid-based strip technology (micro-IDent®plus11, Hain Lifescience, Nehren, Germany) [4]. Identification of the serotype b strains and JP2 genotype strains was performed using the PCR technique [24].

Then, after asking for an additional biofilm sample, cultivation and isolation of the *A. actinomycetemcomitans* strain were performed. After confirming the identification (JP2 genotype), determination of antibiotic resistance to amoxicillin and metronidazole was done using the microbroth-dilution technique. Further, the toxicity to monocytic cells of human origin (MONO-MAC-6; DSMZ no. ACC 124) was assessed. Those were maintained in RPMI 1640 medium containing 10% fetale bovine serum (FBS) and, after washing, adjusted to 10^6^/mL in RPMI 1640. Forty hour cultures of *A. actinomycetemcomitans* strains on agar plates were adjusted to 2 × 10^7^/mL in RPMI 1640. Both suspension were mixed 1:1 and the vitality of MONO-MAC-6 cells was determined after 6 h of incubation at 37 °C with 5% of CO_2_ using the MTT assay, according to Mosmann [44]. As controls, *A. actinomycetemcomitans* HK1651 (JP2 genotype) and *A. actinomycetemcomitans* Y4 (both strains obtained from ATCC, #ATCC 700685, and ATCC 43718), as well as leukotoxin (2.5 µg/mL; purified as described by Kachlany et al. [45] from culture supernatant of a control *A. actinomycetemcomitans* HK1651 strain added by a final centrifugation using a 10 kDa centrifugal filter to remove proteins of lower weights), were used.

## 5. Conclusions

Colonization of Caucasians by the JP2 genotype of *A. actinomycetemcomitans* is rare. In the present study, the microbiological diagnosis played the key role for selecting the use of adjunctive systemic antibiotics, as well as for the ensuring an accurate periodontal diagnosis and adequate treatment for the patient’s teenage daughter.

## Figures and Tables

**Figure 1 pathogens-09-00178-f001:**
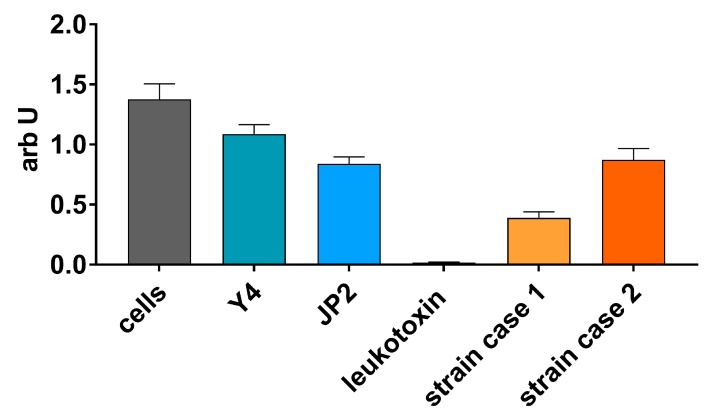
MTT ((3-(4,5-dimethylthiazol-2-yl)-2,5-diphenyltetrazolium bromide) tetrazolium) assay assessing the vitality of the MONO-MAC-6 cells after contact with the clinical *Aggregatibacter actinomycetemcomitans* JP2 genotype strains (case 1 and case 2) in comparison with *Aggregatibacter actinomycetemcomitans* leukotoxin, a reference JP2 genotype strain (HK1651) and Y4 strain.

**Table 1 pathogens-09-00178-t001:** Baseline data. PD, probing depth.

	Patient 1	Patient 2
**Age (years)**	50	55
**Gender**	f	f
**Mean probing depth in mm**	2.5	4.36
**Number of sites ≥ 5 mm PD**	8	43
**Mean attachment loss in mm**	2	5
**Bleeding on probing in %**	47	31
**Plaque index in %**	38	75
**Number of teeth**	26	29

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
