# Peer review of "JP2 Genotype of *Aggregatibacter actinomycetemcomitans* in Caucasian Patients: A Presentation of Two Cases"

_pathogens, 2020, doi:10.3390/pathogens9030178_

Round 1
Reviewer 1 Report
Manuscript ID: pathogens-720686
This is an interesting manuscript about detection and treatment of the JP2 clone of the bacterium Aggregatibacter actinomycetemcomitans. While this clone is frequently detected among individuals of African descent reports of non-African JP2 carriers are scares. However, a few cases are earlier reported. There are probably a couple of JP2 cases which are overlooked in the absence of laboratory diagnostics of this clone.
According to earlier reports of JP2 carriers, both of African and non-African descent, the majority of them are young. Thus, it is valuable to show that also older carriers of the JP2 clone exists.
Since carriers of JP2 are at high risk to develop disease early detection of this clone is important for relevant treatment, including use of antimicrobials as adjuvant. Thus, the summery of the paper is a valuable point.
This reviewer finds the Introduction of the study relevant and satisfactory and have no comments regarding the treatment of the patients.
However, there are some questions and comments regarding the methods and the results.
1) For primarily detection and quantification of A. actinomycetemcomitans the microIDentRPlus11-method was used. When samples were cultivated for isolation purpose was also quantification of A. actinomycetemcomitans performed? If Yes, could a rough correlation between the methods be observed?
2) Was nonJP2 detected in the samplers?
3) Reference for the microbroth -dilution technique is missing.
4) AST of A. actinomycetemcomitans is tricky to perform since no species-specific clinical breakpoints are avaible. For amoxicillin the breakpoint for Haemophilus influence is used (2 mg/L). Breakpoint for metronidazol is set to 4 mg/l for anaerobic bacterial species. It is doubtful if this breakpoint can be used for A. actinomycetemcomitans. Use of amoxicillin and metronidazol is based on clinical studies. Comments regarding this aspect would be valuable.
5) The results of the MTT is confusing. Is the JP2 strain a clinical isolate or a reference strain? The difference in toxicity of this strain and Y4 and strain 2 is rather low. In contrast, the toxicity of the leukotoxin and strain 2 is miore relevant. Is it probably that the toxicity of strain 1 and 2 is related to the carriers? Could it possibly be other reasons? Was A. actinomycetemcomitans from the JP2 positive daughter toxicity-tested?. It is not expected to see such a difference of leukotoxicity between 3 various JP2 isolates. Could this be discussed, or re-examined, to clarify this phenomenon?
Author Response
Here our point to point answers to Reviewer 1. We are very grateful to the highly valuable comments that significantly improved our manuscript.
Reviewer: For primarily detection and quantification of A. actinomycetemcomitans the microIDentRPlus11-method was used. When samples were cultivated for isolation purpose was also quantification of A. actinomycetemcomitans performed? If Yes, could a rough correlation between the methods be observed?
Our response: We found a high number of colonies typical for A. actinomycetemcomitans on the agar plates, about 105 per sample. From studies made before (Eick & Pfister. J Clin Periodontol. 2002;29:638-644, and also unpublished data 2015) we know that the microIDentRPlus11 is highly sensitive and specific and that there is a good accordance between cfu counts and the staining of the bands at the strips of the kit.
Revised text: “Cultivation confirmed the high counts (105) of A. actinomycetemcomitans…” The strain could be also cultured (about 105 per sample) and…
2) Was nonJP2 detected in the samplers?
Our response: We did not detect a non JP2 genotype strain in the samples.
Revised text: “Another A. actinomycetemcomitans (without deletion in the promoter region of the leukotoxin gene was not detected.” and “No other A. actinomycetemcomitans strain was identified.”
3) Reference for the microbroth -dilution technique is missing.
We added a reference.
4) AST of A. actinomycetemcomitans is tricky to perform since no species-specific clinical breakpoints are avaible. For amoxicillin the breakpoint for Haemophilus influencae is used (2 mg/L). Breakpoint for metronidazol is set to 4 mg/l for anaerobic bacterial species. It is doubtful if this breakpoint can be used for A. actinomycetemcomitans. Use of amoxicillin and metronidazol is based on clinical studies. Comments regarding this aspect would be valuable.
Our response. It is right that there are no specific breakpoints for A. actinomycetemcomitans and also we use the MIC values set for Haemophilus influenzae. Further, we screened for beta-lactamase activity which was not added in the manuscript. We replaced the terms “high susceptibility” by “low MIC values”.
We agree that the combination amoxicillin / metronidazole works in clinic. Anyway it, is of interest to determine the MIC values. We comment on that
Revised text: “The in-vitro resistance of the strains to amoxicillin and metronidazole was determined. These data cannot transferred directly to the clinic. A synergism between metronidazole and amoxicillin is well known, as amoxicillin increases the uptake of metronidazole in the bacterial cells [..]. First this combination was used to treat successfully patients with A. actinomycetemcomitans associated periodontitis [..].
5) The results of the MTT is confusing. Is the JP2 strain a clinical isolate or a reference strain? The difference in toxicity of this strain and Y4 and strain 2 is rather low. In contrast, the toxicity of the leukotoxin and strain 2 is more relevant. Is it probably that the toxicity of strain 1 and 2 is related to the carriers? Could it possibly be other reasons? Was A. actinomycetemcomitans from the JP2 positive daughter toxicity-tested?. It is not expected to see such a difference of leukotoxicity between 3 various JP2 isolates. Could this be discussed, or re-examined, to clarify this phenomenon?
Our response:
The JP2 genotype strain as the control was the reference strain (HK1651). We added that information now. The tests were made twice with similar results. The relative low toxicity might be related to the experimental setting. One clinical isolate was absolutely comparable to the reference strain (JP2 genotype). We did not culture the strain of the daughter. However, of interest is the second strain with no contact to Africa.
The JP2 genotype reference strain (HK1651) was more toxic than the Y4 strain. The difference might be not very high but can be related to the experimental conditions with using a lower bacterial concentration and a different cell line than reported before [..]. … Here, we can only speculate if there is a difference in production of leukotoxin or in retaining it at the cell-surface. Leukotoxin is enriched in outer-membrane-like vesicles […]. The obviously very high toxicity may contribute to the infection of a person with no genetic predisposition.
Reviewer 2 Report
This manuscript is a case description of two periodontitis cases both being positive for the highly leukotoxic JP2 clone of Aggregatibacter actinomycetemcomitans (Aa). Aa is considered a key periodontal pathogen. The two females are of Caucasian origin, and 50 and 55 years of age, respectively. The cases are interesting for several reasons, such as being Caucasians whereas most JP2 clone carriers have been reported to originate from North-West Africa. However, one of the cases was married to a husband originating from North Africa, but after microbiological sampling her husband was not found to be positive for Aa. Furthermore, an age predilection for infection with JP2 clone of Aa in young individuals has been suggested, but these two female carriers of the JP2 clone are in the 50’ties. Also, it is mentioned that Porphyromonas gingivalis was not found in bacterial samples from the patients.
Both cases were successfully treated with adjunctive antibiotics, and the JP2 clone of Aa was eliminated.
The present manuscript is addressing a very interesting topic, such as the JP2 clone of Aa being highly associated with rapid and severe development of periodontitis. Over the years, it has been discussed, if microbiological diagnostics is of any value in the treatment of periodontitis. These cases demonstrate examples, where microbiological diagnosis was important in the decision of integrating the use of antibiotics as a part of the treatment plan.
Overall, the cases are of interest, and several reports on the presence of the JP2 clone of Aa and its association with periodontitis have been based on individuals living in the Nordic European countries, whereas fewer reports addressing the topic has been based on patients from Central European countries.
Also, the use and relevance of microbiological diagnostics are addressed and discussed, a topic that has been controversial for many years.
Overall, the discussion is reflecting a great level of knowledge on the topic, and especially the discussion brings up many interesting topics and open questions.
However, when this is said, this manuscript needs to be improved significantly before it is ready for publication.
Some specific points are mentioned below:
The language needs to be improved significantly. Some sentences are unclear and not properly written. Extensive language revision of the whole manuscript is needed. The terminology needs to be improved significantly. Examples: In the title, ‘JP2 strains’ is mentioned, but is ‘JP2 genotypes strains of Aggregatibacter actinomycetemcomitans’ meant? In Fig. 2, what is ‘JPs´?? What is ‘MTT’? Has abbreviation been explained? Is Fig. 2 dealing with Aa strains? It is not mentioned. And several different words are linked with ‘JP2’, e.g., ‘JP2 strains’ , ‘JP2 clone’, ‘JPs’, and ‘JP2-infection (see line 151), ‘JP2-presence’ (see line155) and line 210 just ‘JP2’. Is ‘JP2 clone of Aa’ or JP2 genotype of Aa‘ meant? Please be stricter in the use of the terminology throughout the manuscript. In the title ‘2’ should be ‘two’ (see line 4). Is ‘PDD’ (line 115) the same as ‘PD’? (please see line 15 and many other lines). Line 17: ‘highly toxic’ is it the same as ‘highly leukotoxic’? Line 30: ‘in course of periodontal disease’ is ‘periodontitis’ meant? The Illustrations should be improved significantly. Concerning Fig. 1.: The text and numbers in diagrams are too small and not possible to read. It is suggested that the data in the diagrams are transfers into some kind of table format, where data are more aggregated and described as more overall results for each of the two patients. Concerning Fig. 2: The figure legends in the illustration is insufficient. The Figure legend should be more ‘self-explanatory’ and complete. Line 33, ‘department’ should be ‘Department’ Lines 164-165: The sentence is not clear. ‘The two cases presented here do not display the characteristics of an aggressive periodontitis [33] as……??’ Please, explain further. Line 172: ‘The first patient…’ it could be more clear if you used the age, as the two patients did not have the same age. Lines 222-223: in both patients? jaws? Please, be more precise in the descriptions and wording. ‘… the classification set in 1999…. ’ by Armitage?? ‘…Further, the toxicity to monocytic cells of human origin (MONO-MAC-6; DSMZ no. ACC 124).’ More information is needed. It appears that the sentences is not finalized! And is information on company and country missing? Line 236: As controls, actinomycetemcomitans JP2 and A. actinomycetemcomitans Y4 as well as leukotoxin….’ From where? Culture collection? The layout of the reference list is not finalized and not properly made. A weakness of the case description is that there is no longitudinal perspective, no follow-up results. How long time has the patients been followed until now?Author Response
Comments to Reviewer 2
Reviewer 2
This manuscript is a case description of two periodontitis cases both being positive for the highly leukotoxic JP2 clone of Aggregatibacter actinomycetemcomitans (Aa). Aa is considered a key periodontal pathogen. The two females are of Caucasian origin, and 50 and 55 years of age, respectively. The cases are interesting for several reasons, such as being Caucasians whereas most JP2 clone carriers have been reported to originate from North-West Africa. However, one of the cases was married to a husband originating from North Africa, but after microbiological sampling her husband was not found to be positive for Aa. Furthermore, an age predilection for infection with JP2 clone of Aa in young individuals has been suggested, but these two female carriers of the JP2 clone are in the 50’ties. Also, it is mentioned that Porphyromonas gingivaliswas not found in bacterial samples from the patients. Both cases were successfully treated with adjunctive antibiotics, and the JP2 clone of Aa was eliminated.The present manuscript is addressing a very interesting topic, such as the JP2 clone of Aa being highly associated with rapid and severe development of periodontitis. Over the years, it has been discussed, if microbiological diagnostics is of any value in the treatment of periodontitis. These cases demonstrate examples, where microbiological diagnosis was important in the decision of integrating the use of antibiotics as a part of the treatment plan.
Overall, the cases are of interest, and several reports on the presence of the JP2 clone of Aa and its association with periodontitis have been based on individuals living in the Nordic European countries, whereas fewer reports addressing the topic has been based on patients from Central European countries.
Also, the use and relevance of microbiological diagnostics are addressed and discussed, a topic that has been controversial for many years.
Overall, the discussion is reflecting a great level of knowledge on the topic, and especially the discussion brings up many interesting topics and open questions.
However, when this is said, this manuscript needs to be improved significantly before it is ready for publication.
Some specific points are mentioned below:
The language needs to be improved significantly. Some sentences are unclear and not properly written. Extensive language revision of the whole manuscript is needed. The terminology needs to be improved significantly. Examples: In the title, ‘JP2 strains’ is mentioned, but is ‘JP2 genotypes strains of Aggregatibacter actinomycetemcomitans’ meant?
Our response: we improved the English writing.
In Fig. 2, what is ‘JPs´?? What is ‘MTT’?
Our response: We added more information to the legend.
Has abbreviation been explained? Is Fig. 2 dealing with Aa strains? It is not mentioned. And several different words are linked with ‘JP2’, e.g., ‘JP2 strains’ , ‘JP2 clone’, ‘JPs’, and ‘JP2-infection (see line 151), ‘JP2-presence’ (see line155) and line 210 just ‘JP2’. Is ‘JP2 clone of Aa’ or JP2 genotype of Aa‘ meant? Please be stricter in the use of the terminology throughout the manuscript.
Our response: We checked the manuscript thoroughly and corrected accordingly.
In the title ‘2’ should be ‘two’ (see line 4). Is ‘PDD’ (line 115) the same as ‘PD’? (please see line 15 and many other lines).
Our response: We agree with the reviewer and used the term PD “probing depth” throughout the manuscript.
Line 17: ‘highly toxic’ is it the same as ‘highly leukotoxic’?
Our response: we agree with the reviewer and used only “highly leukotoxic”
Line 30: ‘in course of periodontal disease’ is ‘periodontitis’ meant?
The Illustrations should be improved significantly. Concerning Fig. 1.: The text and numbers in diagrams are too small and not possible to read. It is suggested that the data in the diagrams are transfers into some kind of table format, where data are more aggregated and described as more overall results for each of the two patients.
Our response: we are grateful to this valuable comment and used a table to display the data.
Table 1: Baseline data
|
|
Patient 1 |
Patient 2 |
|
Age (years) |
50 |
55 |
|
Gender |
f |
f |
|
Mean probing depth in mm |
2.5 |
4.36 |
|
Number of sites ³5 mm PD |
8 |
43 |
|
Mean attachment loss in mm |
2 |
5 |
|
Bleeding on probing in % |
47 |
31 |
|
Plaque index in % |
38 |
75 |
|
Number of teeth |
26 |
29 |
Concerning Fig. 2: The figure legends in the illustration is insufficient. The Figure legend should be more ‘self-explanatory’ and complete.
Our response: We modified the legend “Figure 2. MTT ((3-(4,5-dimethylthiazol-2-yl)-2,5-diphenyltetrazolium bromide) tetrazolium) assay assessing the vitality of the MONO-MAC-6 cells after contact with the clinical Aggregatibacter actinomycetemcomitans JP2 genotype strains (case 1 and case 2) in comparison to Aggregatibacter actinomycetemcomitans leukotoxin , a reference JP2 genotype strain (HK1651) and Y4 strain
Line 33, ‘department’ should be ‘Department’
Our response: We corrected this mistake.
Lines 164-165: The sentence is not clear. ‘The two cases presented here do not display the characteristics of an aggressive periodontitis [33] as……??’ Please, explain further.
Our response: We agree with the reviewer and changed the respective sentence to:
In contrast, both patients presented here were at the time of the baseline examination between 50 and 55 years of age and were diagnosed with chronic periodontitis.
Line 172: ‘The first patient…’ it could be more clear if you used the age, as the two patients did not have the same age.
Our response: We agree with the reviewer and included the age for the patients.
Lines 222-223: in both patients? jaws? Please, be more precise in the descriptions and wording. ‘… the classification set in 1999…. ’ by Armitage?? ‘…
Our response: The diagnosis of chronic periodontitis was established for both patients. According to the Armitage classification no distinction between the localization of the defects is made. We therefore added: by Armitage
Further, the toxicity to monocytic cells of human origin (MONO-MAC-6; DSMZ no. ACC 124).’ More information is needed. It appears that the sentences is not finalized! And is information on company and country missing?
Our response: The cells were bought at the German Collection od Microorganism and Cell Cultures GmbH which is expressed by the given number DSMZ no. ACC 124. We restructured the sentence. Further, the toxicity to monocytic cells of human origin (MONO-MAC-6; DSMZ no. ACC 124) was assessed
Line 236: As controls, actinomycetemcomitans JP2 and A. actinomycetemcomitans Y4 as well as leukotoxin….’ From where? Culture collection?
The origin of the strain was added. Both were obtained from ATCC. Leuktotoxin was purified in our laboratory, the reference is given.
The layout of the reference list is not finalized and not properly made.
Our response: we improved the reference list and used the recommended download for endnote.
A weakness of the case description is that there is no longitudinal perspective, no follow-up results. How long time has the patients been followed until now?
Our response: the patients went back to private practice, this was added to the text. The 50-year-old remained for another 6 months at our Department before she was sent back to her dentist. Whereas the 55-year-old patient remained in the recall at our Department for two more years after the active therapy.
Round 2
Reviewer 1 Report
The authors have responded satisfactory to the reviewers comments.
However, some additional comments/suggestions are here mentioned.
1) The title of the paper has been changed, by a mistake I guess. Should be: JP2 genotype of Aggregatibacter actinomycetemcomitans in Caucasian patients : a presentation of two cases.
2) line 90: replace leukotoxin gene by leukotoxin operon
3) line 148: mispelling of JP2 genotype
4) line 262: delete strain. (strains are collected from plates, i.e. individuals do not colonized by strains. Suggestion: Colonization of Caucasians by the JP2 genotype of Aggregatibacter actinomycetemcomitans is rare).
5) The JP2 genotype is considered to be more leukotoxic than the non JP2 genotype. Thus, it is surprising that the difference in leukotoxicity between Y4 and HK1651/strain 2 is limited. Difference in leukotoxicity among JP2 strains have been reported by Birkehol-Jensen and co-authors (Differental Cell lysis among periodontal strains of JP2 and non JP2, 2019). Could be included as further information regarding differences in leukotoxicity among JP2 strains.
Author Response
Dear reviewer
We are grateful for the valuable comments and contribution to our manuscript. We followed the succinct advice and revised our manuscript accordingly. See below our point-to-point answers.
Yours faithfully,
Alexandra Stähli
1) The title of the paper has been changed, by a mistake I guess. Should be: JP2 genotype of Aggregatibacter actinomycetemcomitans in Caucasian patients : a presentation of two cases.
Our response: we thank the reviewer for their meticulous reading and changed the title.
2) line 90: replace leukotoxin gene by leukotoxin operon
Our response: we replaced leuktoxin gene by leukotoxin operon
3) line 148: mispelling of JP2 genotype
Our response: we corrected this typo.
4) line 262: delete strain. (strains are collected from plates, i.e. individuals do not colonized by strains. Suggestion: Colonization of Caucasians by the JP2 genotype of Aggregatibacter actinomycetemcomitans is rare).
Our response: we followed the reviewer`s suggestion and wrote: “Colonization of Caucasians by the JP2 genotype of A. actinomycetemcomitans is rare.”
5) The JP2 genotype is considered to be more leukotoxic than the non JP2 genotype. Thus, it is surprising that the difference in leukotoxicity between Y4 and HK1651/strain 2 is limited. Difference in leukotoxicity among JP2 strains have been reported by Birkehol-Jensen and co-authors (Differental Cell lysis among periodontal strains of JP2 and non JP2, 2019). Could be included as further information regarding differences in leukotoxicity among JP2 strains.
Our response: we are grateful for this comment and added this information to the discussion line 244: “Here, it has to be pointed out that leukotoxicity may be different depending on the test method that was used [38]. Variation of leukotoxicity was not only observed among JP2- and non-JP2 genotype of A. actinomycetemcomitans but also among the methods Western blotting, ELISA, cell lysis assay, and mRNA expression assay [38].“
Reviewer 2 Report
There is still a need for language editing, just by looking at the abstract.For example, the title is not properly written.
Words as 'leukotoxic' and 'leucotoxic' are used - I expect and recommend to choose one way to write the word throughout the manuscript.
Abstract, Line 13: Is the word 'origin' missing?
Abstract, Line 14: Is the word 'presence' missing?
The sentences in the abstract should be more precise and completely written. The text is not easy to read.
Comma is not used properly, making the text more difficult to read.
Just be reading the abstract again, the paper needs to be more clearly written to make the content clear and possible to read and understand.
Author Response
Dear reviewer
We would like to thank the reviewers for their thoughtful and constructive comments regarding our manuscript. We have addressed all the comments brought forth. All changes were incorporated into the revised manuscript and are highlighted in the manuscript text file.
Yours faithfully,
Alexandra Stähli
Point-to-point answers for Reviewer 2
Words as 'leukotoxic' and 'leucotoxic' are used - I expect and recommend to choose one way to write the word throughout the manuscript.
Our response: we corrected this inconsistency throughout the manuscript.
Abstract, Line 13: Is the word 'origin' missing?
Our response: we added the word “origin”
Abstract, Line 14: Is the word 'presence' missing?
Our response: to our understanding no word is missing
The sentences in the abstract should be more precise and completely written. The text is not easy to read.
Our response: we simplified some sentences in the abstract to make it easier for the reader.
Comma is not used properly, making the text more difficult to read.
Our reponse: we checked the commas
Just by reading the abstract again, the paper needs to be more clearly written to make the content clear and possible to read and understand.
Our response: we proofread the whole manuscript and tried to transmit the content as clearly as for us possible.